# Senior Immigrant Women and Participation in Voluntary Services in Norway

**DOI:** 10.3390/healthcare11152162

**Published:** 2023-07-29

**Authors:** Rakel Bruland, Wenche Malmedal, Lene Blekken

**Affiliations:** Department of Public Health and Nursing, Norwegian University of Science and Technology, 7491 Trondheim, Norway; rakelb_1996@hotmail.com

**Keywords:** senior immigrant women, voluntary services, participation, community, language learning, health, qualitative study

## Abstract

In Norway, there is a rising trend in the number of senior immigrants. Engagement in society is known to be crucial to their well-being and health, with participation in voluntary activities being one way of partaking in a community. A qualitative study was conducted, involving 12 immigrant women aged 50 years and above. Semi-structured interviews were utilized to gain insight into their experiences of participation in activities, aiming to identify both facilitators and barriers to their engagement in activities provided by voluntary organizations. The analysis resulted in two main themes and five sub-themes. According to the women’s experience, learning Norwegian was important for successful integration. They also emphasized the importance of social relationships for improved self-perceived health and a sense of belonging in a community. Language challenges and limited social integration were barriers negatively impacting their participation, with health being a crucial factor determining their capacity to engage in voluntary services. Since most of the women talked about their health, this became an important topic of further investigation. This article is a contribution towards gaining further insight into the experiences of senior immigrant women participating in voluntary services and explores the challenges faced by voluntary organizations in improving information dissemination and minimizing barriers to participation. Public health competence can improve knowledge regarding facilitating health-promoting participation in voluntary services.

## 1. Introduction

Norwegian national policies have stood out due to their emphasis on ensuring equality, universal services and solidarity, contributing to comprehensive policies to reduce social health inequalities [1]. Public health work should promote the population’s well-being, health, good environmental and social conditions, and help to prevent somatic and mental illnesses, injuries, or disorders [2]. In Norway, the Public Health Act was adopted in 2011 and implemented on 1 January 2012. The purpose of the act is to contribute to societal development that reduces social inequalities in health and promotes public health, while also facilitating systematic and long-term public health work [2].

Dahlgren and Whitehead [3] address the social determinants of health, necessitating actions at multiple levels of society and dealing with complex political considerations concerning resource distribution among various social groups. The focus on healthy lifestyles aligns with traditional public health efforts, and studies indicate that the local Norwegian health promotion continues to prioritize this approach [2]. The Public Health Act includes the idea of ‘’Health in All Policies’’, where health promotion becomes the responsibility of all sectors, instead of relying exclusively on the health sector. This necessitates systematic assessments of the health impact of all policies and actions [4]. The implementation of ‘’Health in All Policies’’ demands sustainable effort, a long-term vision, and an emphasis on inter-sectorial collaboration, participation, and information sharing [5,6]. The public health work includes reaching all immigrants with a health-promoting perspective, where disease prevention, reducing social inequalities, and strategies for preserving health are essential [7]. In Norway, the Integration Act can help reduce social inequalities, as its purpose is to contribute to a good understanding of Norwegian society and provide Norwegian language education. Immigrants between the ages of 18 and 67 who have been granted a residence permit have an obligation and a right to participate in Norwegian language education and Norwegian social studies [8].

In general, immigrant seniors who are physically and socially active are in a better condition to lead meaningful lives and better integrate into society [9]. Healthier aging is a benefit for the individual and for society as it strengthens the self-mastery of the individual and contributes to reducing and postponing the need for health care services [10].

Through the quality reform “A full life—all your life—A Quality Reform for Older Persons,” the Norwegian government came up with subsidy schemes for physical and social activities in order to increase participation among seniors [11]. The reform builds on what health care personnel, seniors, families, voluntary organizations, researchers, and leaders have found to work in practice [11].

Health inequality exists between the immigrant population and the general population, whereas the immigrant population experiences a higher occurrence of reduced social networks and social support, which increases the risk of poor health conditions [12,13]. Immigration affects the social determinants of health [14], and senior immigrant women are especially susceptible to social exclusion, which can exacerbate feelings of loneliness and contribute to mental health challenges [12]. Negative experiences early in life can lead to poor health in older years [3], and there is a lack of knowledge about whether immigrants experience better quality of life after moving to Norway [12]. Health challenges before, during, and after emigration; language barriers; physical inactivity; and economic conditions are some examples of influencing factors related to health that should be examined more closely [12]. The process of migration often entails leaving behind friends, countries of birth, and social communities, resulting in a sense of loss that includes shared language, previous roles, and societal functions [12]. When individuals lack the feeling of being recognized or valued in their social environment, the experience of loneliness can increase [15]. To address these issues, endeavors aimed at enhancing psychosocial relationships, such as establishing meeting places, encouraging community involvement, and fostering social networks, should increasingly encompass senior immigrants [12]. Social need is depicted in the desire to belong to a group. We humans want to add value to ourselves, others, and the community. The theory of mattering comprises two complementary psychological experiences: adding value and feeling valued. Human beings can add value to and feel valued by work, a community, other people, and themselves. Mattering affects health by making us feel appreciated, respected, and recognized [10]. Sense of belonging has proved to be health-promoting and vital for maintaining overall well-being, particularly with increased age when one may experience changes in health [12]. Nevertheless, senior immigrants may experience challenges in establishing a sense of belonging, especially in the presence of conflicting identities and cultural contrasts [16]. The absence of a sense of belonging can give rise to loneliness and reduced well-being, particularly in situations where coordinating meaningful social interactions and shared understanding is challenging [17].

The impact of community participation and well-being is influenced by various demographic and environmental factors, including age, employment, and genetic resilience [3]. Thus, it is crucial to address the issue of health diversity, particularly when advocating effective international policies on equality [3]. From a public health perspective, stimulating the community participation of immigrants is expected to achieve positive health gains [12,13]. Voluntary organizations play an important role for community participation [18], and the Public Health Act emphasizes the importance of collaboration between the voluntary sector and municipalities [2].

Services provided by voluntary organizations facilitate meeting places and participation in social activities, in which it is free of charge to participate [10,11]. The absence of meeting places can create conditions for mental and physical challenges. Therefore, participation in voluntary activities can be an important social gathering place [19]. Voluntary organizations offer the opportunity to reach different groups of residents, contributes with valuable insights, and functions as a source of knowledge and ideas that the municipality can benefit from [20]. Furthermore, previous empirical studies have demonstrated that voluntary organizations are considered a central arena for integration [21].

Caritas is an example of an international voluntary organization doing humanitarian work and is one of the largest help organizations, located in 200 countries worldwide. Among other things, Caritas offers language learning, physical activities, computer learning, and social events. In Norwegian society, voluntary organizations act as supplements to public offers [10,19]. Voluntary organizations, in addition to municipal services, can also contribute to health promotion efforts, language learning, and increased activity levels among senior immigrants [22]. The activities offered by the voluntary organizations should benefit society and enable contributions to society, and voluntary organizations facilitate this by helping to create meeting places in the local area and thereby ensure community participation, which can be essential for engaging seniors [10,11].

Studies have demonstrated that social participation with an increased age is positively associated with a higher level of activity of daily living. However, this association appears to vary based on the participants’ gender, their age, and the types of activities [23]. Voluntary services exhibit a social gradient, with disadvantaged groups being less likely to engage in voluntary activities. Recognizing voluntary services as a public health strategy can yield potential benefits for all parties involved in addressing health inequalities and social exclusion [24]. There is inequality in participation in voluntary services, where it is especially common among individuals with higher education and income, good health, a large social network, and those integrated into the local community [25,26,27,28,29,30]. Statistics Norway (SSB) provides data on civic engagement, categorized by gender and age. The statistics reveal a higher level of active involvement among men in Norway, compared to women. Additionally, there is a greater representation of men as members of different organizations. Moreover, participation in organizations decreases by almost 10 percent from the age range of 45–66 years to 67 years and older [31].

There are several voluntary organizations in Norway that offers activities specifically to participants with an immigrant background. Some of these organizations work towards increased participation in physical activity among immigrants through exercise groups for women [32]. Such voluntary activities can reduce the number of individuals at risk of lifestyle-related diseases [33]. Nevertheless, Jakobsen and Spilker [32] found that senior immigrant women utilize exercise facilities and other public health measures to a lesser extent than the general population. Gele and Harsløf [34] point out the lack of knowledge and information about available voluntary services, limited trust in the community, and language difficulties as barriers to participation among senior immigrants. This supports the findings of Egge [35] and Horne [36], who emphasize that trust and language comprehension are important for participation and inclusion.

Since voluntary organizations facilitate a healthy and fair community while ensuring an equilibrium among values required to ensure the experience of mattering, the Norwegian government wishes to collaborate with voluntary services to mobilize a common effort to prevent loneliness and contribute to social support [10,37]. Collaboration between voluntary services and municipalities is therefore important for public health work [2], as voluntary services create places for society to meet and facilitate community participation.

Previous empirical studies have demonstrated that voluntary organizations contribute to health benefits for participants and facilitating community inclusion by offering language learning [38,39]. Participating in voluntary activities can benefit senior immigrants by enhancing their cultural and social capital, facilitating their integration into the host society. Moreover, participation in a community is considered a crucial determinant in the health and well-being of seniors [35].

## 2. The Present Study

Norway is increasingly characterized by diversity, with a projected rise in the number of immigrants in the coming years [9]. Understanding the desires of senior immigrant women is crucial from a public health perspective, as it enables voluntary services to offer relevant, health-promoting programs and community participation. Many seniors have communicated being presented with limited opportunities to adequately partake in a variety of activities [11]. As a response, there are examples of systematic efforts that municipalities have implemented to expand the availability of different activities via collaboration alongside voluntary organizations and the local community to create joyful events for seniors [11]. Given the heightened vulnerability of senior immigrant women to social exclusion, their experiences can contribute to mental health challenges [12]. Therefore, incorporating this sample into the study can provide an innovative aspect by exploring the experiences of senior immigrant women in engaging with voluntary services, discussing both facilitators and barriers to their involvement in activities provided by voluntary organizations. These findings can enhance our understanding of how voluntary services can promote participation, not only in Norway but also in other countries. The present study aims to further explore senior immigrant women’s experiences of participating in voluntary services, and discover barriers and facilitators to engagement in activities offered by voluntary organizations. Thus, the research question is as follows:

How do senior immigrant women experience participation in voluntary services?

## 3. Methods

The present study used a qualitative design based on a phenomenological and hermeneutic approach [40]. Through individual interviews, data were gathered to explore the subjective experiences from the perspective of senior immigrants. The interviews with the women were analyzed to gain an in-depth understanding of their individual experiences in participating in voluntary services.

### 3.1. Data Collection

Individual, semi-structured interviews were utilized to collect data for this study [40]. The interviews were initiated and completed between September 2021 and November 2021. All interviews were conducted individually with the participants by the first author (FA). FA is a female student with a Master’s Degree. FA acquired fundamental proficiency in conducting interviews from various subjects through her educational journey. Due to language challenges, all participants were offered a publicly approved interpreter who would be present during the interviews. Eight of the twelve participants accepted this offer. An interview guide with open-ended questions, provided by FA, was used. Furthermore, a welcoming atmosphere was made to encourage the participants to express themselves openly (see Table 1). In addition, to facilitate in-depth answers and reflections on the themes in the interview guide, the interviewer took a narrative approach in asking the participants to present concrete episodes and examples from their own life [41]. During the interviews, FA also summarized and repeated the participants’ expressions, asking for confirmation on the accuracy of their perceptions. The interviews took place at the voluntary service center at times that suited the participants and were audio-recorded and transcribed verbatim. Each interview lasted 20–40 min. The interviews were not extensive, with relatively short answers from the participants. To facilitate theoretical saturation, logs were written after conversations with individuals who had participated in voluntary activities organized by Caritas. After the completion of the last activities lead by the first author, there was a summary with the participants discussing what went well and what could be conducted differently regarding voluntary activities. However, the logs from the conversations were not utilized because what the participants said did not provide further understanding of the shared information during the interviews. This indicates that despite the short interviews, the sample size was sufficient to comprehend the phenomenon and adequately address the research questions, as Thagaard [41] suggests is important in qualitative research.

As the FA conducted the interviews, many preconceptions were challenged upon receiving the participants’ answers to the presented questions. The participants’ responses developed new knowledge about the significance of voluntary services. What the FA learned about the participants’ experiences provided new knowledge that helped refine their own preconceptions. The FA asked new questions that yielded new answers, which created a deeper understanding of the theme.

### 3.2. Participants and Context

The participants were selected by purposive sampling and recruited from the Caritas voluntary service center in a city in Norway where they participated in various types of voluntary activities, such as language learning, data learning, meetings in cafés, and physical activities. FA contacted the local center, and a recruitment procedure was planned together with one of the leaders. To become familiar with the informant group and to facilitate recruitment, FA offered to do some volunteer work at the center. During autumn 2021, FA held two lectures about physical activity and health, and for a period of three months, she held weekly Aerobic/Zumba exercise classes for immigrant women at the center. FA also took part in several other activities at the center. While participating in the work of the center, FA gave information about the research project and asked the women face-to-face about participating in the study. The women interested in participating contacted FA and they agreed upon a time and date for the interview. Five participants were recruited from the exercise classes, and seven were recruited after being giving information about the research project during other activities at the center.

In this study, immigrants are defined as persons who immigrate to Norway, were born abroad from foreign parents, and have four foreign grandparents [42]. In Norway, the State Senior Council has defined everyone over 50 years as seniors and distinguishes between younger and older seniors [43]. The inclusion criteria were age ≥50 years, females, and immigration from another country. Being socially stigmatized by virtue of being a senior, women and immigrants can together conduce to increased health problems and illness in the Nordic countries among immigrant women [44]. To investigate this further, the inclusion criteria in this study comprised senior immigrant women.

### 3.3. Analysis and Interpretation

The method for analyzing and interpreting the transcribed text followed the principles of Kvale and Brinkman [40]. The authors utilize a data-driven inductive analysis approach based on hermeneutics and phenomenology, employing a phenomenological analysis termed ‘’meaning condensation’’ consisting of five steps in this study. Firstly, all authors read the transcribed texts carefully several times to establish an overall picture. Second, meaning units were decided as they appeared from the interviews, and essential statements and quotes were identified. At this stage, it was important to let the text speak for itself by letting the meaning from the interviews come forward without interpretation. Step three entailed looking for common themes and sub-themes among the meaning units. When deciding on the theme and sub-theme, it was important to find a balance between keeping close to the data and the viewpoint of the participants, as well as the understanding and interpretation of the researchers. As for the fourth step, the meaning units were analyzed based on the research question and specific aim of the study. The fifth step entailed presenting the results, and the key main findings have been summarized in a descriptive statement. The process of interpreting the text was circular, involving a continuous movement between specific parts of the text and the text as a whole [40]. All authors discussed the descriptions and textual interpretations and then reached a consensus on the themes and sub-themes describing senior immigrant women’s experiences of participating in voluntary services and its perceived impact on their health. Table 2 shows examples of quotes, meaning units, sub-themes, and themes.

### 3.4. Ethics

This study received ethical approval from the Norwegian Centre for Research Data (NSD), Registration no: 746132. The participants in the study were recruited after receiving information about the research project during activities at the center. A total of 15 women were interested in participating in the project. Every participant received both written and oral information about the study, and written informed consent was collected from each participant. Furthermore, the participants were also assured that they could terminate the interview at any point without the need for explanation or providing a reason. There were three women who wanted to join but withdrew when the interview started, and this resulted in a total number of 12 participants. Because these women withdrew without providing any justification, FA did not ask why. The interviewer emphasized that this was entirely acceptable. The publicly approved interpreter signed a project-specific non-disclosure agreement. To ensure the anonymity of all participants, all identifiable characteristics are excluded from the data presentation. The women were informed that they had the opportunity to review their own transcribed interview, provide comments, and verify its accuracy. No participants provided feedback on the transcribed interview or findings.

## 4. Results

A group of 12 senior immigrant women were interviewed, whose ages ranged from 53 to 79 years. Table 3 shows the demographic characteristics of the study participants.

The results section is categorized by themes and sub-themes from the analysis.

The analysis resulted in two main themes and five sub-themes. Theme 1 was health and physical activity, with its sub-themes being (a) the importance of physical activity and (b) poor health conditions. Theme 2 was being part of a community, with the sub-themes being (a) learning Norwegian, (b) sharing thoughts and experiences, and (c) barriers to participation in a community.

### 4.1. Health and Physical Activity

The immigrant women expressed that physical activity was important and that their perceived health affected their participation in voluntary services. This study will present the importance of physical activity and various experiences associated with poor health.

### 4.2. Importance of Physical Activity

Most women reported that participating in voluntary services, specifically physical activities, resulted in health benefits. It was important to many participants to be active through voluntary services or social engagements by planning social walks or other physical activities to decrease time spent home alone. Socializing and being physically active together with other people were important, and participants reported better mental health when meeting others and participating in activities: “We are sitting home and depressed, people need to go out and join activities … be together as friends, then mental health gets better … When you are alone, you forget things, a lot of things, a lot of important things, then you need to talk and be active with other people” (informant F). Other women reported different ways physical activities affected their health. In particular, women who lived alone and had no family in Norway emphasized the importance of getting out: “If you are home all the time, you get stressed … when you are joining other people and sharing training it gets better, then you are finished, go home and rest” (informant A).

Some women reported that they were telling other people about voluntary activities and, by doing so, recruiting them. “My neighbor was four years at home, no activities or physical activities at all. When I came, she walks with me every day, every day. That is good for mental health” [informant E]. The desire to contribute and tell others about language learning and being physically active was important to many participants: “They who are like me or older, they have to participate in physical activity and other services, and they have to share, not only look at information and be home” [informant B]. Many participants said it was important to tell others about voluntary services to avoid someone sitting at home and missing out.

### 4.3. Poor Health Conditions

Most of the women shared information about their health because this affected their participation in voluntary services. All participants reported poor physical health conditions as being back or knee pain. Some participants reported that they were trying to decrease their activity level to prevent further aggravation of the pain. “Poor health conditions make it hard to participate in physical activities … Now I have health issues, so I do not participate in activities as much as before” (informant G). Other participants reported that they would increase their activity level to prevent pain. Some women reported that they were active before they experienced poor health conditions, with a desire to become more active again in the future. Most women believed that their poor health conditions affected their participation in voluntary services either by not participating due to bodily pain or because they wanted to prevent pain and health issues by making health-promoting choices.

Experiencing poor health conditions due to feelings of missing family and being lonely was consistent among senior immigrant women living alone. Specifically, several women with family living abroad often felt anxious and stressed. These women reported that spending most of their time at home increased their stress levels and negatively affected their health. Among women who lived alone, voluntary participation was an important part of their everyday life. When the participants met other people through voluntary activities, they were distracted from their loneliness and stress for a while. Spending time together with other people and sharing physical activities in groups led to better experiences of health and helped them feel more connected with others: “Physical activities are good for me. I am just sitting home and stressing a lot. Physical activities or training makes things better” (informant K).

### 4.4. Being Part of a Community

All the women found that learning Norwegian was crucial for successful integration and a sense of belonging. At the same time, some women experienced different challenges when it came to learning the language and participating in voluntary services. The collected data illustrates various ways in which the women experienced learning the Norwegian language and participating in a community.

### 4.5. Learning Norwegian: The Key to Integration

All the participants mentioned the importance of learning the Norwegian language and wanted increased language proficiency through participation in voluntary Norwegian education. The women reported that Norwegian education was more important to them than physical activity: “Learning Norwegian and participating is more important than activities” (informant G). Some women reported that they would rather work instead of joining physical activities because of the opportunity to learn Norwegian: “I like work because I want to speak more Norwegian. Now I do not speak much, not good … I’m stressed … no Norwegian” (informant J).

Some women pointed out how physical activity could facilitate Norwegian learning through communication with other participants and the instructor. If the instructor was Norwegian and explained the movements, then the participants could learn the words while being active. The participants reported that it could be better to learn Norwegian through hearing words and doing certain movements rather than learning the language at school: “If I get an offer like that, it is better than just being at school” (informant B). Some of the women had completed adult language education at school and felt that they had forgotten what they had learned.

Some participants preferred voluntary activities that focused on both learning Norwegian and physical activities for senior immigrants: “Some activities where we talk, because then we get Norwegian in our lives. I was going to school five years, one year ago I started to forget. That is why activities where we talk and use the language are important” (informant F); “women my age like dancing … dancing activities where we get together, talk and use the language … Seniors have a hard time learning languages, whereas it would be fun with activities where older people can use this language” (informant G). Most of the women were interested in increasing their amount of physical activity if they could learn Norwegian at the same time. When asked about speaking Norwegian during physical activities, one informant said, “It is important for me. I can hear and understand and talk a little bit of Norwegian” (informant I). The women also mentioned that Norwegian was important in order to communicate and process information regarding voluntary activities.

### 4.6. Sharing Thoughts and Experiences

Most of the senior immigrant women believed that sense of community was a main factor for participation in voluntary services. Moreover, many women experienced positive emotions from feelings of belonging in a community and reported that the most important part of the activities was the social support from meeting other people that they provided because they experienced social connectedness and felt valued.

Some participants fled from war. For these women, physical activity and social relations were the most important. Some women emphasized that meeting in groups, exercising, and talking about the concerns they had was a positive experience. “It is good for the sense of community to talk together about problems … I am concerned for my family and their lives. I think there should be group work, that there are activities and opportunities to tell others about our own problems. Then there is less to hide inside” (informant F). The voluntary activities made it possible to meet one group of people many times, with a focus on social relations. By being active together in a group, many women experienced a feeling of happiness and unity.

### 4.7. Barriers to Participating in a Community

Some participants avoided voluntary activities because of the barriers from not knowing other participants. “I get extra stressed and nervous when I’m not familiar with everyone and if I can’t find my way to the activity” (informant F). Women who lived alone spent more time explaining barriers to participating in voluntary activities compared to those who had someone they knew in the area. The women without family and friends in the area reported that they felt insecure because they did not know anyone else attending the activities. Uncertainty regarding where the voluntary activities were to be held, as well as economic barriers, were critical reasons for not participating. Fear of getting lost and not getting to the activities were also factors that reduced participation: “I like it, but at the same time not so much because I cannot always find my way and I feel stupid” (informant K). Several women mentioned the economic barriers regarding bus passes. If they had a bus pass, they would join voluntary activities more often: “If I had bus pass, then I would participate more” (informant L). Many women lived far away from where the activities took place and were thus dependent on public transportation.

## 5. Discussion

This study aimed to investigate further into the experiences of senior immigrant women participating in voluntary services, uncovering both facilitators and barriers to engagement in activities provided by voluntary organizations. The participants encountered various gains and barriers during their involvement in voluntary activities. Additionally, the women expressed the consequence of learning Norwegian for successful integration and the importance of social relationships for improved self-perceived health and a sense of community. Since most of the women talked about their health, this became an important topic to investigate further.

### 5.1. Health

The present study demonstrates that senior immigrant women who engaged in voluntary physical activities experienced positive effects on health and a reduced sense of stress. Studies support the idea that physical activities can contribute to maintaining a good quality of life, health, physical function, and reducing the risk of falling [46,47,48,49]. In general, health is one of the most crucial factors encouraging voluntary participation among senior people [50]. Being physically active and taking part in social engagements were important to the women not only in decreasing time spent home alone but also in meeting other people and participating in activities. Other studies have demonstrated that gender, social status, and ethnicity taken together can explain the negative effects on one’s health. Being socially stigmatized by virtue of being a senior, immigrant, and of a lower socioeconomic status all at once may lead to heightened illness and health challenges among immigrant women [44]. Voluntary organizations can positively influence these factors by providing a venue for physical and social activities. After participating in such voluntary activities, the women in this study reported a feeling of well-being and a sense of community. This is supported by Prilleltensky [10], who points out that when community members have a sense of mattering, they are more likely to experience positive relations, physical well-being, and other positive health benefits.

Several participants attempted to increase their activity levels to prevent the further worsening of their pain, which supports the understanding that regular physical activity can help maintain or even improve physical functions as people age [9]. The model of health determinants by Dahlgren and Whitehead [51] indicate that individual life habits affect health. Furthermore, physical activity is one of the most important lifestyle factors from a public health perspective [52].

On the other hand, several participants tried to reduce their activity level to prevent pain, and they reported being more active prior to experiencing health problems. A lack of physical and social engagement leading to isolation among senior immigrant women could be an underlying factor that has resulted in the reported poor health conditions. The fact that senior immigrant women exercise less than the population at large [33] might indicate that this group is more exposed to lifestyle diseases due to inactivity. However, they encountered discomfort and barriers that prevented them from taking part. This dilemma, called the prevention paradox [53], presents a quandary where participants potentially benefiting the most from voluntary services may believe that the offer has the least effect. Some women experienced poor health conditions due to feelings of loneliness, especially senior immigrant women who lived alone and missed their family. Prilleltensky [10] points out that isolation and loneliness are related to higher mortality rates and increased risk of heart attacks and cognitive decline among seniors. Voluntary services can impact factors related to physical activity and loneliness by offering a wide range of activities. Therefore, it is important to reduce barriers to participation so that everyone gets the chance to engage in health-promoting activities.

### 5.2. Community

Most of the women experienced positive feelings when socializing with the other participants. A feeling of unity replaced some of the worries and challenges in their everyday life when participating in voluntary activities. Social communities are health-promoting and contribute to well-being. Activity, participation, and socialization through a community are all important prerequisites to achieving good health and functioning [54]. We know that many seniors are lonely and isolated, which has a negative impact on health and functioning [54]. This study demonstrates that talking to fellow participants further increased their interest in voluntary services. Thus, the phenomenon of mattering contributes to a healthy community [10], and this finding also corresponds with what Meld. St. 19 [55] and Schrempft et al. [56] suggest, that is, when communicating effectively and achieving a feeling of mutual understanding in a social environment, we can experience increased well-being and better health.

Several of the women in this study reported that meeting other women made them feel less lonely. This phenomenon is supported by Dahlberg et al. [17], who claim that meeting others in a social community reduces the risk of loneliness. Since loneliness affects senior immigrants twice as much as the general population [11], it might be important for this group to participate in activities together with others. The senior immigrant women wanted to inform others about voluntary activities because they believed that other people could also benefit from participating, and they reported that receiving new information from others helped them feel connected and valued. The theory regarding mattering [10] supports this development, pointing out that adding value to and feeling valued by the community, oneself, and others is important in relation to self-perceived health. Equally, Whitehead and Dahlgren [3] point to the importance of social networks for good health and well-being; indeed, the women expressed a desire to extend information about physical activity and language training to other senior immigrants. This study suggests that social differences exist between senior immigrant women who have connections in the local community and those who do not, which, according to Dahlgren and Whitehead [57], results in inequalities in health. Other studies also suggest that greater social isolation among senior women is linked to higher levels of sedentary behavior and decreased everyday physical activity. Additionally, differences in physical activity can be linked to a higher risk of reduced well-being and poor health in relation to isolation [56].

### 5.3. The Importance of Learning Languages

Language learning is a topic of priority because a common language is the link in social interaction and an important factor for integration into a community [34]. This study indicates that language learning is among the most important factors for successful integration. The women found that the Norwegian language established social relationships and could provide mutual confirmation and shared experiences with other participants. Accordingly, Dahlgren and Whitehead [51] emphasize the importance of a social network as an important social determinant of health. At the same time, the women experienced more challenges learning a new language when older, and several studies support their experiences on this matter [3,17,34,57,58]. Moreover, several women experienced a lack of follow-up on their Norwegian language skills, and those who did not fully understand Norwegian reported feeling ashamed and insecure about communicating in the language. Due to the limitations imposed by the Integration Act, which provides the obligation and right for immigrants under 67 years old to participate in Norwegian language training [8], senior immigrants face reduced support and opportunities. Furthermore, voluntary organizations can be essential in providing Norwegian language training and facilitating inclusion within a community. Collaboration between voluntary organizations and the municipality is crucial in anticipating challenges posed by an increasing number of elderly individuals, higher life expectancy, and changes in diseases [11]. Therefore, conducting studies that investigate the experiences of voluntary services and participation becomes especially important for this group.

Additionally, meeting new societal needs requires innovative activities. Since the municipality holds responsibility for public health in the population, it should recognize the importance of local voluntary organizations and possess knowledge about the potential and characteristics of voluntary services in public health efforts. Collaborating with voluntary organizations allows the municipality to gain a better understanding of public health, as voluntary services involve people working with people. This enables voluntary services to be present where actions and changes occur, providing valuable information to other sectors of society regarding necessary efforts and areas of focus [19].

A lack of ability to express oneself through the language used by the local community can cause a feeling of shame, and when a participant’s mother tongue is no longer widespread in the local community, they may, as Barstad and Molstad [58] argue, not experience communicating like they are used to and therefore feel as if they are not being perceived as they wish. This corresponds to Whitehead and Dahlgren’s claim that people who experience social exclusion in their local environment are more likely to feel ashamed, insecure, and further exposed to lifestyle diseases [56]. Because immigration affect the social determinants of health [14], Whitehead and Dahlgren [3] recommend the creation of meeting places such as voluntary services to facilitate social interaction, in the effort to prevent health challenges. Such meeting places could offer opportunities to learn languages and engage in physical activities that may enhance social relationships. Participating in voluntary activities can also provide individuals with a mutual experience of feeling valued by others [6].

## 6. Methodical Considerations

There are some weaknesses regarding this study. The use of an interpreter in the interviews may have influenced the information provided by the participants, and linguistic challenges may have limited the depth of the interviews. On the other side, the recruitment of women from various countries and age range, from a minority of society being socially stigmatized by virtue of being a senior, woman and immigrant, are the primary advantage of the study. Despite the relatively small sample size, we observed that saturation was attained after only ten interviews, and no further data emerged from the last two interviews. Qualitative studies seek the individual participant’s life experience that is relevant to the research question and are therefore difficult to generalize. How and when to generalize findings from one participant to another is challenging, and a diverse sample should be sought [40]. The study sample is too narrow for the findings to have general validity. However, the study findings can deepen other studies on the same topic and provide additional dimensions.

Previous studies have indicated that 16 or fewer interviews should be sufficient to achieve data saturation in qualitative studies [59]. The crucial aspect in evaluating the project is therefore not whether an exact number of participants or cases is specified, but whether the researcher has based their sampling strategy on well-founded reasoning. However, the feasibility of the study can be assessed based on the project description, which should estimate the approximate amount of data to be collected. A high number of participants carries a greater risk of poor scientific quality compared to a low number of participants [60].

Not all participants are directly quoted in the results, but all participants’ experiences are included in the analysis of the findings. The quotes did not fit to be directly extracted from the interviews with all participants. However, emphasis has been placed on presenting the experiences of all participants, and several of the senior immigrant women shared similar experiences on the topics addressed in this study.

All participants in the study are women who were involved in the voluntary organization Caritas, which may have contributed to bias, as the strengths and barriers may not be the same in other voluntary organizations. It would have been interesting to investigate whether senior immigrant women experienced the same benefits and challenges in other voluntary organizations.

Our aim was to enhance the transparency of the analytical approach by providing a comprehensive description of the methodology, thereby increasing the study’s credibility. Furthermore, all findings underwent rigorous discussion within the research group, consisting of seasoned researchers with extensive research experience. Therefore, this enhances the trustworthiness of our findings and the credibility of the research. However, the replication of the study in other national and international contexts would provide a broader picture of senior immigrant women and their experiences related to health, community, and language learning and the role of voluntary services.

### Implications for Clinical Practice and Research

From a public health perspective, it is a public responsibility to provide health-promoting activities in the local community [2], since a lack of social support and participation can lead to poor health among the immigrant population [12]. Caritas is an example of a voluntary organization that can impact health, sense of community, and language learning by offering various activities. If voluntary offers are to be socially viable, it is important to facilitate offers in which everyone can participate. In the case of such efforts, it is important to consider the prevention paradox, where participants who would potentially benefit the most from voluntary services may believe that the offer has the least effect [53], and thus ensure accessibility for those who need the services the most.

Expertise is needed at both the individual and group level to make it possible for everyone, regardless of their functional abilities, to participate in a community and in the society. The public health worker has the knowledge to facilitate participation with the aim to increase individuals’ control over factors that affect their health [61]. Public health work can operate at a group level by implementing measures for specific target groups such as senior immigrant women experiencing health challenges. Meanwhile, at an individual level, public health work can offer counseling and health information for older immigrant women in general and those with health problems in particular. Public health workers have the expertise to facilitate the participation of both those who do and do not face health challenges in activities [61]. Thus, the municipality and public health workers should collaborate with voluntary organizations to facilitate opportunities for residents to feel part of a community. Conducting qualitative research on the perspectives of municipal employees regarding integration among senior immigrants in public health initiatives would therefore be a valuable area of investigation. To fully understand the prevalence of and risk factors for poor health conditions among senior immigrant women, further quantitative research is necessary.

## 7. Conclusions

This study indicates that senior immigrant women’s experiences of participation in voluntary services may be positively influenced by the presence of supportive networks and culturally sensitive programming. The women’s self-perceived health was influenced by their participation in the community. Furthermore, their involvement in the community affected the extent to which they listen, speak, and learn the language. This, in turn, influenced the amount of communication with other individuals and how the women experienced a sense of belonging, well-being and health. Barriers, such as language challenges and limited social integration, would negatively impact their participation, and health was a crucial factor that determined the capacity for participation in voluntary services. Senior immigrant women forwarded information about voluntary services to others and wanted to be a part of a community and add value to society. Based on the women’s experiences, it seems that health, sense of community, and language learning are mutually dependent on each other. Strengthening public health competence could enhance knowledge about facilitating health-promoting participation in voluntary services.

## Figures and Tables

**Table 1 healthcare-11-02162-t001:** Information about main themes from the interview guide.

Can you tell me about yourself? Thoughts about physical activity. Can you tell me about your experiences with physical activity? Contributions as a user of physical activity services. What do you believe could encourage other immigrant women to participate in physical activity?

**Table 2 healthcare-11-02162-t002:** Examples of interpretations.

Quotes	Meaning Units	Sub-Themes	Themes
When I feel stressed, I find that physical activity is very important. If you stay at home, you get sick. It is good for the brain and body to get outside. Poor health conditions make it hard to participate in physical activities. Physical activities are good for me. I am just sitting at home and stress a lot. Physical activities or training makes things better.	The participants found that physical activity is important for physical and mental well-being. Health problems were a barrier to participation in voluntary services. For some women, it was important to participate in voluntary activities to reduce poor health.	Importance of physical activity Poor health conditions.	Health and physical activity.
Learning Norwegian is more important than activities. I have to speak Norwegian … language is very important. That we are friends and go out together improves our mental health … we can reminisce, talk, we can be active. It is good for the sense of belonging to talk about problems together. I get extra stressed and nervous when I do not know everyone and if I cannot find my way to the activity. If I had a bus pass, then I would participate more.	Experiences regarding the importance of being able to speak and understand Norwegian are important. The women wanted to focus more on Norwegian language learning. Most women want to go to voluntary activities to avoid being home alone and to share thoughts with others. The participants found that it is good to meet and talk with other participants. The activities offered become a platform for meeting other people and getting out of the house. Fear of not finding one’s way and not knowing any other participants are barriers to participation. Economic barriers such as transportation and distance were crucial barriers to participation.	Learning Norwegian is the key to participation in a community. Sharing thoughts and experiences. Barriers to participation in a community.	Being part of a community.

**Table 3 healthcare-11-02162-t003:** Characteristics of the study sample.

Participant	Age	Continent of Birth	Years Lived in Norway	Education ^1^	Interpreter Included in Interview
A	64	Africa	3 years and 8 months	Adult education	Yes
B	53	Africa	7 years	Adult education	Yes
C	56	Africa	11 years	-	Yes
D	65	Asia	3 years and 6 months	Adult education	Yes
E	58	Asia	8 years	Completed adult education	Yes
F	60	Asia	6 years	Completed adult education	Yes
G	72	Asia	6 years	Adult education	Yes
H	58	Africa	8 years	-	No
I	57	Asia	9 years	Adult education	No
J	70	Asia	5 years	Voluntary teacher	No
K	79	Africa	12 years	-	Yes
L	60	Africa	22 years	Health worker	No

^1^ The Integration Act can help reduce social inequalities, as its purpose is to contribute to a good understanding of Norwegian society and provide Norwegian language education [8]. Norwegian adult education is offered by municipalities throughout the country and is based on the principle that good language skills are crucial for participation in the community, and engaging in social and cultural activities. The goal of adult education in Norway is to provide participants with the necessary language and cultural skills to integrate into Norwegian society, and participate in the community. Adult education is regulated by the Adult Education Act, whereas the provision of courses is the responsibility of the respective public education authorities at the various levels of education [45].

## Data Availability

The data that support the findings of this study are available from the corresponding author upon reasonable request.

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
