# Peer review of "Senior Immigrant Women and Participation in Voluntary Services in Norway"

_healthcare, 2023, doi:10.3390/healthcare11152162_

Round 1

Reviewer 1 Report

1. We need a specific explanation of Norwegian socio-cultural background as to why Norwegian migrant women (over 50) are socially branded and restricted to social activities. For example, in the case of countries that value Confucian culture, this phenomenon can occur differently. In addition, in some countries that implement immigration policies, this phenomenon may appear differently.

2. Therefore, of course, it is necessary to interpret the content in the discussion.

3. Why is volunteering especially important in Norway during participation in various social life? Is there any other way of participating in social life?

4. Please match the format of the qualitative research in the journal Healthcare

5. Please add information about COREQ. 

6. The specifics of COREQ are easy to find on Google.

7. In qualitative inquiries, interviews can only be stopped if they are reached to the theoretical saturation rather than the number or time. Please add a description of this part.

1. We need a specific explanation of Norwegian socio-cultural background as to why Norwegian migrant women (over 50) are socially branded and restricted to social activities. For example, in the case of countries that value Confucian culture, this phenomenon can occur differently. In addition, in some countries that implement immigration policies, this phenomenon may appear differently.

2. Therefore, of course, it is necessary to interpret the content in the discussion.

3. Why is volunteering especially important in Norway during participation in various social life? Is there any other way of participating in social life?

4. Please match the format of the qualitative research in the journal Healthcare

5. Please add information about COREQ. 

6. The specifics of COREQ are easy to find on Google.

7. In qualitative inquiries, interviews can only be stopped if they are reached to the theoretical saturation rather than the number or time. Please add a description of this part.

Reviewer 2 Report

Review report for healthcare-2492375

This is a relevant and interesting qualitative study aiming to explore the factors that enhance and prevent immigrant women’s engagement in voluntary activities in Norway. The rationale, the method, and the analysis of the data seem to be sound. Thus, I have reason to believe that the study has merit and should be further considered for publication pending some revisions that I outline below.

1.      The introduction would benefit from a brief discussion of engagement in voluntary activities as way to improve well-being (happiness, life satisfaction, quality of life). I feel this is an important issue that has been neglected.

2.      The introductory section feels quite under-developed to me. For instance, there is an absence of literature review on previous empirical quantitative or qualitative studies on the topic of civic engagement/ engagement with voluntary activities.

3.      In addition to the above, I am also not convinced about the innovation of the study. Specifically, the authors should explain more what this study contributes to the international literature by reviewing evidence both within and beyond the context of the current study.

4.      Given the age of the sample, the authors should also engage in a discussion regarding the connection between civic engagement and ageing. There is quite a lot of work on this topic (see for example: https://doi.org/10.1016/B978-0-12-380880-6.00016-2 and https://doi.org/10.1093/geront/48.3.368).

5.      Perhaps, the “mattering” term (line 54) could be replaced with meaning/purpose in life since “mattering” sounds odd?

6.      Immigrants are particularly vulnerable to be socially excluded or face mental health difficulties. Therefore, working with this sample should be discussed as an innovate aspect of the current study.

7.      Furthermore, in the introduction, it is unclear what is the role of the gender in civic engagement. The authors could review some relevant studies describing gender differences in civic engagement.

8.      The section “purpose” from line 68-73 is very short and should be expanded to include what are the gaps in knowledge or evidence base this study seeks to address. This section could be re-titled as “the present study”.

9.      Some good information on the types of civic engagement and, specifically, voluntary activities are provided in a section entitled “introduction” (lines 74-101); however, this should be part of the general introductory section. Thus, there should not be a distinction between “Background” (line 32) and “Introduction” (line 74).

10.   Please explain a little bit more what adult education refers to. Is this a stage before higher education?

11.   Line 111: what does RRB stand for?

12.   The table 2 title could be better rephrased as “codebook” or “examples of interpretations”, perhaps. As the title is now written, it does not make much sense.

13.   I am wondering whether the analysis of discursive texts (transcripts) could be better explained as thematic content analysis?

14.   The terms “topics” could be better phrased as “themes”. This is more typical of qualitative study.

15.   In general, the description of the method and results sections is very well done!

16.   The authors ought to discuss more from a theoretical perspective the concept of thematic saturation (line 425). The authors can discuss what have previous studies found regarding the number of interviews required for thematic and coding saturation in qualitative studies.

17.   I definitely think that the authors should discuss how the sampling of all interviewees from a single voluntary organization, namely Caritas, could have biased the results.

I appreciate the authors’ work and hope that my comments would help with further refining this manuscript.

Reviewer 3 Report

Afer reviewing I found these issues:

1. Even the manuscript has a background and an introduction, the hypothesis of the work is not well defined. In addition, the authors used multiple self senences in these two parts with no references (exp: line 43, 75, 83...). the ideas are also not well organized (exp: backgrond: the authors begin with a definition from a decree and later they explain that the decree was adptedin 2011.

I suggest to associate the two parts, organize and reorder the ideas, and define the hypothesis.

2. In the methods, the authors did not explain how the participant were reruited and where they were found. The authos should also include what was the total number of individuals contacted (line 173-174). 

2. The results are not well organized. you should begin with the population characteristics and then explain your results. Yet all is mixed in your results making them dificult to understand. Also the first part of the results: "lines 179-189" repeated exactly what was shown in the objecteives. Delete please.

The same remak for lines 184-190 and line 200-201: try to avoid repitition and orgnize your ideas.

It is not clear what was the total number since some f them where no included in te interpretaion (table 3). why they were not exactly included in te analysis? why were they included in the results? explain please. 

On the other hand, why was the education missing for some participants (especially if we know that there is a low number of participants and having missing data for a number of 8 or 12 participants is not suitable).

In table 3: the education (1) legend should be added as a foot note just below the table.

Also, you shoumd review the numbers of th titles : exp: Line 202 : the number should be 5.1.1 not 5.2.......correct all the subsequent titles please

The same for the discussion: in adition to the fact that its too long (which needs to be somewhat summarized), the titles should be the same wih the results with the adapted numbers.

You should also add a conclusion and extend the limitations of the study.

I have also some "minor " coments:

1. Add the name of the country i n the title

2. Line 111: define RRB

3. Line 146: correct "and"

4. Separate subtopics "importance of physical activity" from the column of "meaning units".

5. I don't think that figure 1 is really interesting. I suggest to delete it.

6. Delete the colouring from the references

Round 2

Reviewer 2 Report

The authors have done exceptional work in revising the manuscript according to the reviewers' comment. I am satisfied with the responses to the issues I raised in the first round of review. Thus, I have no further comments. 

I wish the authors a speedy publication!

Reviewer 3 Report

I would thank the authors for their efforts to improve the quality of the manuscript; however, I still have some concerns:

The introduction is too long, and was not sufficiently summarized (even the titles of were changed). In addition, the ideas are still not well organized an fail to define the need of the work (Why was the study conducted, previous studies in this field..).

The discussion was also not summarized (it was extended).

Even the conclusion was reformulated it did not provide sufficient data regarding the main results (it contains only  4 lines (619-623), after some definitions (another introduction)

I have also other remarks

Abstract:

The abstract is not well organized : it contains an introduction of 3 lines, methods (12 lines), results (3 lines) and a conclusion (6 lines) which is not adapted. The results should be extended and the methods should be summarized.

Line 66-97: Justify the text

Line 94: correct the reference style

Line 104: correct the reference 10

Line 159: I did not understand how did you pass from reference 39 to ref 55. Be careful and verify all your references.

Methods : it is not necessary to include the name of the authors in the methods

Results:

Delete sentence of line 286-87

 You should begin with the 303-304 (and provide more details about the demographics)

Delete the color of line 536
